# Hybrid-MST: A Hybrid Active Sampling Strategy for Pairwise Preference Aggregation

**Jing Li**
LS2N/IPI Lab
University of Nantes
jingli.univ@gmail.com

**Rafal K. Mantiuk**
Computer Laboratory
University of Cambridge
rkm38@cam.ac.uk

**Junle Wang**
Turing Lab
Tencent Games
wangjunle@gmail.com

**Suiyi Ling, Patrick Le Callet**
LS2N/IPI Lab
University of Nantes
suiyi.ling, patrick.lecallet@univ-nantes.fr

## Abstract

In this paper we present a hybrid active sampling strategy for pairwise *preference* aggregation, which aims at recovering the underlying rating of the test candidates from sparse and noisy pairwise labelling. Our method employs Bayesian optimization framework and Bradley-Terry model to construct the utility function, then to obtain the Expected Information Gain (EIG) of each pair. For computational efficiency, Gaussian-Hermite quadrature is used for estimation of EIG. In this work, a hybrid active sampling strategy is proposed, either using Global Maximum (GM) EIG sampling or Minimum Spanning Tree (MST) sampling in each trial, which is determined by the test budget. The proposed method has been validated on both simulated and real-world datasets, where it shows higher preference aggregation ability than the state-of-the-art methods.

## 1 Introduction

Preference aggregation from annotators' pairwise labeling on the test candidates is a traditional but still active research topic. As the name implies, the objective of preference aggregation is to infer the underlying rating or ranking of the test candidates according to annotator's (users or players) binary label, e.g. which one is better? In particular, recently, with the access of big data, preference aggregation from pairwise labeling has been widely applied in recommendation systems such as on movie, music, news, books, research articles, restaurant, products according to user's preference selection; or in social networks for aggregating social opinions; or in sports race, chess and online games to infer the global ranking of the players, etc.

In some applications, such as game players matching systems (e.g. MSR's TrueSkill system[1]), friends-making website and subjective image/video quality assessment (IQA/VQA) [2], discovering the underlying scores of the test candidates is more important than the rank order so the system could know the intensity of the preference from users, eventually to assign matching players to the on-line game players, or recommend the possible friends who have the same interests to the users, or to quantitatively evaluate the performance of different coding/rendering/display techniques in IQA/VQA domain. However, as the size of the test candidates $n$ gets bigger, which is happening nowadays, the number of required pairwise labeling grows exponentially $O(n^2)$ leading to the unfeasible implementation. Thus, there is an urgent need to reduce the number of pairwise comparisons, that is, selecting part of the pairs but without loosing the aggregation accuracy.

In this paper, we present a hybrid active sampling strategy for pairwise labeling based on Bradley-Terry (BT) model[3], which can convert pairwise preference data to scale values. **This work considers not only about inferring ranking but also recovering the underlying rating**. The term *Hybrid* explains that different sampling strategies are used in this method determined by the test budget. Active learning recipe is adopted in our strategy by maximizing the information gain according to Lindley's Bayesian optimal framework[4]. To capture the latent rating information, the minimum spanning tree (MST) is employed where the pairwise comparison is considered as a undirected graph. The MST guarantees the strong connection and eventually leads to higher prediction precision by BT model. In addition, the MST allows for a parallel implementation on pairwise comparison through crowdsourcing platform (such as Amazon MTurk), i.e. multiple annotators could work at the same time. Source code is public available in Github [1].

The main contributions of our work are highlighted as follows: **1) Batch mode facility**: When the number of test candidates is $n$, the proposed Hybrid-MST active sampling strategy allows for $n-1$ parallel pairwise comparison each time. **2) Erroneous tolerance**: We didn't model annotator's behavior in this work, however, the utilization of MST to some extent tolerates the malicious labeling from spammers (who give wrong/random answers). **3) Low computational complexity**: Compared to the state-of-the-art method that considers numerous parameters and deals with both active sampling and noise removing (e.g. Crowd-BT [5]), Hybrid-MST has much less time complexity. **4) Application flexibility**: Hybrid-MST is applicable in all conditions where aggregation on ranking or rating or both is required. It is also conductible in both small-scale lab test environment or large-scale crowdsourcing platform.

The remainder of this paper is organized as follows. State-of-the-art work is introduced in Section 2. The proposed Hybrid-MST strategy is presented in Section 3 containing both theoretical analysis and Monte Carlo simulation analysis. Extensive experimental validation on simulated dataset and real-world datasets are shown in Section 4. Finally, Section 5 concludes this work.

## 2   Related Work

In real applications of preference aggregation, annotator's label could be **explicit**, for instance, a Likert scale score from "excellent" to "bad", or **implicit**, e.g. pairwise comparison voting on two test candidates. The explicit label is more likely to be inconsistent [6][7] and noisy due to diverse influence factors [8]. According to a well known phenomenon in psychological study of human choice that "human response to comparison questions is more stable in the sense that it is not easily affected by irrelevant alternatives"[9], obtaining label from pairwise comparison is thus a more appealing way for human participated labeling application, such as image quality assessment. Nevertheless, in whatever types of pairwise comparison, pairwise labeling still suffers from noises from a variety of sources, such as the human annotator's expertise, the emotional states of players in a match, or the environment (external factors) of competition venue. In such case, the challenge changes to how to invert this implicit and in most cases noisy pairwise data back to the true global ranking or rating.

Several models have been proposed to explain the relation between pairwise-comparison responses and ranking/rating scale, including the earlier heuristic methods Borda Count[10], and the currently widely used probabilistic permutation model such as the Plackett-Luce (PL) model[11][12], the Mallows model [13], the Bradley-Terry (BT) model[3], and the Thurstone-Mosteller (TM) model[14]. When facing the large-scale data but with sparse labels, these models might have computational complexity issues or parameter estimating issues. Thus, in machine learning community, numerous studies have been focusing on optimizing the parameters of these models[15][16], designing efficient algorithms [17][18], providing sharp minimax bounds [19] and proposing novel aggregation models[9][20][21]. Meanwhile, some researches are aiming at develop novel models to infer the latent scores of the test candidates from pairwise data and eventually obtain the rank ordering[6] [22][23][24].

It is well known that pairwise comparison needs large number of pairwise data to infer the ranking, which is in most applications very time consuming. A straightforward way to boost the pairwise labeling procedure is through data sampling. A simple and straightforward pair sampling strategy is random sampling such as the "balanced sub-set" method proposed by Dykstra [25] by putting the test candidates in a form (triangle, or rectangular matrix) only subsets of the test candidates are

compared, and the HRRG (HodgeRank on Random Graph) method proposed by Xu *et al.* [26] where random graph is utilized and only connected vertices are compared, meanwhile a Hodge theory based rank model (HodgeRank) is proposed to convert the sparse pairwise data to scale ratings. Another way to sample pairs is based on empirical observations that comparing closer/similar pairs would be more important than the distant pairs. In [27], the authors proposed to apply the sorting algorithms to sample pairs. In [28][29], Li *et al.* proposed an Adaptive Rectangular Design (ARD) to adaptively and iteratively selecting pairs based on the estimated rank ordering of test candidates.

To further improve the aggregation performance, the recent studies focused on active learning for information retrieval. In [30], the authors exploit the underlying low-dimensional Euclidean space of the data to discover the ranking using a small number of pairwise comparisons. Some other researches focus on selecting the pairs which could generate the maximum information gain defined by a utility function. In [31], the sampling strategy is based on TM model by employing the Bayesian optimization framework, while Chen et.al. [5] (Crowd-BT) utilizes the BT model but also considers the annotator's influence. Xu *et al.* [32] (Hodge-active) employs the HodgeRank model as well as the Bayesian information maximization to actively select the pair.

Active learning based sampling methods have demonstrated their outstanding performance in different datasets. However, they still have at least one of the following drawbacks: 1) The sampling procedure is a sequential decision process, which means the generation of next pair is determined only when the previous observation is finished. Such sequential mode is not suitable for large-scale (e.g. crowdsourcing) experiments, in which many conditions are tested in parallel. 2) Most of the proposed methods focus on ranking aggregation, which might not be accurate enough for the applications that require ratings scores. 3) Annotator's unreliability on labeling the pairwise data should be considered in the active learning process, in other words, the active sampling strategy should be robust to observation errors. A straightforward way is to model annotator's behavior, as done for the Crowd-BT method [5]. However, it is computationally expensive.

To resolve the challenges mentioned above, in this paper, we proposed a hybrid active sampling strategy which allows for batch mode labeling and be robust to annotator's random/inverse labeling behavior to infer the scale ratings. Details are introduced in the following sections.

## 3 Proposed Methodology

Let us assume that we have $n$ objects $A_1, A_2, ...A_n$ to test in a pairwise comparison experiment. The underlying quality scores of these objects are $\mathbf{s} = (s_1, s_2, ...s_n)$. In an experiment, the annotator's observed score for object $A_i$ is $r_i$. $r_i$ is a random variable $r_i = s_i + \epsilon_i$, where the noise term is a Gaussian random variable $\epsilon_i \sim \mathcal{N}(0, \sigma_i^2)$. In a single trial, if $r_i > r_j$, then the annotator selects $A_i$ over $A_j$, and the outcome is registered as $y_{ij} = 1$. If $r_i < r_j$, then $y_{ij} = 0$. For the case that $r_i = r_j$, $y_{ij}$ is randomly assigned with 0 or 1 (In real test, the annotators in such condition could randomly make a selection). The probability of selecting $A_i$ over $A_j$ is denoted as $Pr(A_i \succ A_j)$.

### 3.1 Preference aggregation model

There are already some well-known models to convert the pairwise probability data to cardinal scale ratings as we mentioned before. In this study, we choose BT model as an example. But this work could be easily extended to generalized linear model (GLM), in which BT model is the *logit* condition, and TM model is the *probit* condition.

According to BT model, for any two objects $A_i$ and $A_j$, the probability that $A_i$ is preferred over $A_j$, i.e. $Pr(A_i \succ A_j)$ could be represented as:

$$Pr(A_i \succ A_j) \triangleq \pi_{ij} = \frac{\pi_i}{\pi_i + \pi_j}, \qquad \pi_i \geq 0, \qquad \sum_{i=1}^{t} \pi_i = 1 \tag{1}$$

where $\pi_i$ is the *merit* of the object $A_i$. The relationship between underlying score $s_i$ and $\pi_i$ is $s_i = \log(\pi_i)$, thus, we obtain:

$$\pi_{ij} = \frac{e^{s_i}}{e^{s_i} + e^{s_j}} = \frac{1}{1 + e^{-(s_i - s_j)}} \tag{2}$$

Since we measured is a distance value between two objects, there are in total $n - 1$ free parameters that need to be estimated. To infer the $n - 1$ parameters in BT model, the Maximum Likelihood

Estimation (MLE) method is adopted in this study. Given the pairwise comparison results arranged in a matrix $\mathbf{M} = (m_{ij})_{n \times n}$, where $m_{ij}$ represents the total number of trial outcomes $A_i \succ A_j$, the likelihood function takes the shape:

$$L(\mathbf{s}|\mathbf{M}) = \prod_{i<j} \pi_{ij}^{m_{ij}} (1 - \pi_{ij})^{m_{ji}} \tag{3}$$

Replacing $\pi_{ij}$ by $\frac{1}{1+e^{-(s_i - s_j)}}$, and maximizing the log likelihood function $logL(\mathbf{s}|\mathbf{M})$, we could obtain the MLEs $\hat{\mathbf{s}} = (\hat{s_1}, \hat{s_2}, ..., \hat{s_n})$. Generally, there is no closed-form solution for MLEs and they are found numerically. The MLEs $\hat{\mathbf{s}}$ follow a multivariate Gaussian distribution. The covariance matrix $\hat{\Sigma}$ could be estimated using the Hessian matrix of the $logL$ [33]. Thus, for a given pairwise observation $\mathbf{M}$, we could obtain the approximated prior information on $\mathbf{s} \sim \mathcal{N}(\hat{\mathbf{s}}, \hat{\Sigma})$.

## 3.2 Active learning

The purpose of active learning is to gain information from the observations. For a given prior information, the selection of next pair or pairs should provide the maximum information than others. A utility function is thus defined to measure this expected information gain (EIG). Generally, the Kullback-Leibler divergence (KLD) between the prior distribution and the posterior distribution on $\mathbf{s}$ is used as the utility function [5] [31]. Different from them, in this study, we utilize the local pair distribution information rather than the global multivariate distribution to calculate the EIG.

According to the MLEs based on current observations, $\mathbf{s} \sim \mathcal{N}(\hat{\mathbf{s}}, \hat{\Sigma})$. For a pair $\{A_i, A_j\}$, the score distance between them is $s_{ij} \sim \mathcal{N}(\hat{s_i} - \hat{s_j}, \sigma_{ij}^2)$, where $\sigma_{ij}^2 = \hat{\Sigma}(i,i) + \hat{\Sigma}(j,j) - 2\hat{\Sigma}(i,j)$. The EIG of pair $\{A_i, A_j\}$ is defined as the expected KLD between the prior distribution and the posterior distribution of $s_{ij}$, that is:

$$U_{ij} = \int \sum_{y_{ij}} log \left\{ \frac{p(s_{ij}|y_{ij})}{p(s_{ij})} \right\} p(s_{ij}|y_{ij}) p(y_{ij}) ds_{ij} \tag{4}$$

where $p(s_{ij})$ is the prior density, $p(s_{ij}|y_{ij})$ is the posterior density given outcomes $y_{ij}$ ($y_{ij} = 1$ if $A_i \succ A_j$, otherwise, $y_{ij} = 0$). According to Bayes' theorem, $p(s_{ij}|y_{ij})/p(s_{ij}) = p(y_{ij}|s_{ij})/p(y_{ij})$, Equation (4) could be rewritten as:

$$U_{ij} = \int \sum_{y_{ij}} log \left\{ \frac{p(y_{ij}|s_{ij})}{p(y_{ij})} \right\} p(y_{ij}|s_{ij}) p(s_{ij}) ds_{ij} \tag{5}$$

where $p(y_{ij}|s_{ij})$ is the conditional probability density for the outcome $y_{ij}$ in condition $s_{ij}$. We define $p(y_{ij} = 1|s_{ij}) = p_{ij}$, and $p(y_{ij} = 0|s_{ij}) = q_{ij}$, thus, we have $p_{ij} = \frac{1}{1+e^{-s_{ij}}}$, $q_{ij} = 1 - p_{ij}$. The Equation (5) could be rewritten in a tractable computation form :

$$U_{ij} = E(p_{ij}log(p_{ij})) + E(q_{ij}log(q_{ij})) - E(p_{ij})logE(p_{ij}) - E(q_{ij})logE(q_{ij}) \tag{6}$$

where $E(\cdot)$ is the expectation taken w.r.t prior distribution, i.e. $\mathcal{N}(\hat{s_i} - \hat{s_j}, \sigma_{ij}^2)$ . For instance, the first item in Equation (6) could be written in the form:

$$E(p_{ij}log(p_{ij})) = \int p_{ij}log(p_{ij})p(s_{ij})ds_{ij} = \int \frac{1}{1+e^{-x}}log(\frac{1}{1+e^{-x}})\frac{1}{\sqrt{2\pi}\sigma_{ij}}e^{-\frac{(x-(\hat{s_i}-\hat{s_j}))^2}{2\sigma_{ij}^2}} dx \tag{7}$$

This form allows us to use Gaussian-Hermite quadrature [34] for approximation which reduces the computational complexity dramatically. In our study, 30 sample points are used for estimation. An example of the contour plot and mesh-grid plot for the $U$ under different means and standard deviation conditions is shown in Figure 1. According to this figure, the pairs which have similar scores or the score differences have high uncertainties would generate high information, which is consistent with the studies in [27] [28].

## 3.3 Hybrid pair selection strategy

Now, based on the current observations, we could estimate the EIG for all pairs. The next step is to study how to select the pair/pairs based on the EIG.

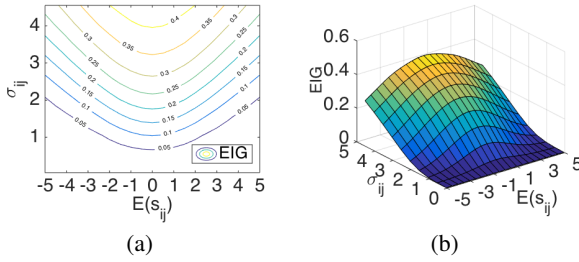

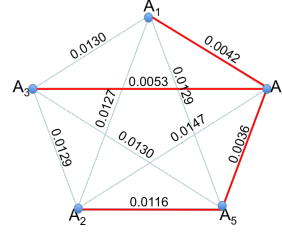

Figure 2: An undirected weighted graph and its MST (red edges).

Figure 1: (a) Contour plot and (b) mesh-grid plot for the EIG in function of $E(s_{ij})$ and the standard deviations $\sigma_{ij}$.

### 3.3.1 Global Maximum (GM) method

A conventional way of active sampling is to select the pair which provides the highest EIG [5][31][32][35], that is:

$$\{A_i, A_j\} = argmax_{i \neq j} U_{ij} \tag{8}$$

However, as we already discussed before, it is a sequential sampling strategy which has limitations in real application such as in large-scale data processing or crowdsourcing platform where parallel execution is necessary. Thus, a method which allows for batch-mode implementation is considered.

### 3.3.2 Minimum Spanning Tree (MST) method

Pairwise comparison could be considered as a undirected graph $G = (V, E)$, where vertices $V$ represent test candidates, and edges $E$ represent whether or not the pairs are compared. In our study, $V = A$, $E = \{(A_i, A_j) : A_i, A_j \in V, m_{ij} + m_{ji} > 0\}$. $w(E)$ are the weights on the edges, in our study, they are the inverse of the EIG of candidate pairs, i.e. $w(E) = \frac{1}{U_{ij}}$.

A MST $G_{mst}$ is a subset of the edges of a connected, edge-weighted (un)directed graph that connects all the vertices together, without any cycles and with the minimum possible total edge weight. The characteristics of MST include:

- If there are $n$ vertices in the graph, then each spanning tree has $n - 1$ edges.
- If each edge has a distinct weight, then there will be only one, unique MST.
- If the weights are positive, then a MST is a minimum-cost subgraph connecting all vertices.

Thus, MST facilitates the batch mode in real application, the strong connection over all test candidates and the maximum sum of information gains of all possible pairs. The pair selection criterion based on MST method is:

$$\{A_i, A_j\} = \{E_{mst} \mid G_{mst} = (A, E_{mst})\} \tag{9}$$

In this study, we use Prim's algorithm [36] to find the MST as it is optimal for dense graphs. An example of an undirected weighted graph and its MST is shown in Figure 2.

### 3.3.3 Threshold setting

In this section we analyze the performance of the GM and MST methods. Firstly, in GM method, we initialize the pair comparison matrix $\mathbf{M}$ by $m_{ij} = m_{ji} = 1, i \neq j$ to fix the resolving issue of BT model [5]. Then, we design a Monte Carlo simulation experiment, assuming 10, 16, 20 and 40 test objects. The underlying scores are uniformly distributed from 1 to 5, with noise $\epsilon_i \sim \mathcal{N}(0, \sigma_i^2)$, $\sigma_i$ is uniformly distributed between 0 and 0.7. In a simulated test, if the sampled score $r_i$ is higher than $r_j$, then $A_i$ is selected over $A_j$. We also model the observation errors that might happen in the real test, i.e. the subject makes a mistake (inverting the vote) during the test. The probabilities of observation errors are designed as 10%, 20%, 30% and 40%. Therefore, there are in total 16 simulated tests, each test repeats 100 times.

To evaluate the aggregation performance of GM and MST, the Pearson Linear Correlation Coefficient (PLCC) and Kendall's tau coefficient (Kendall) between the designed ground truth scores and the MLE scores obtained by BT model are calculated. For easier illustration, in the following section, we define **1 standard trial number** as the total number of comparisons that one observer needs to compare in Full Pair Comparison (FPC), that is, for $n$ objects, 1 standard trial number equals to $n(n-1)/2$ comparisons.

By running Student's t-test on the performances of GM and MST methods and checking their significant difference (which one is better), we find that generally, the GM method performs better than the MST method when the standard trial number is less than 1. With the increase of the comparison numbers, the MST method performs better than GM method, especially when the observation errors are large.

To benefit from both GM and MST methods, we decide to develop a hybrid active sampling strategy with **1 standard trial number as the switching threshold**, i.e.:

$$\{A_i, A_j\} = \begin{cases} argmax_{i \neq j} U_{ij} & \text{if } \sum_{i,j} m_{ij} \leq \frac{n(n-1)}{2} \\ E_{mst} & \text{otherwise} \end{cases} \tag{10}$$

The whole Hybrid-MST sampling strategy is summed up in Algorithm 1.

---

**Algorithm 1** Hybrid-MST sampling algorithm

---

**Input:** Current pairwise observation matrix **M**, Number of test objects $n$
**Output:** Pairs for next round $\{A_i, A_j\}$
  **for** all possible pairs $\{A_i, A_j\}$, $i < j$ **do**
    Computing EIG $U_{ij}$ according to Equation 6
    **if** $\sum_{i,j} m_{ij} \leq \frac{n(n-1)}{2}$ **then**
      Select the pair $\{A_i, A_j\}$ which has the maximum $U_{ij}$
    **else**
      Find MST according to $U_{ij}$, for all $i < j$;
      Select the pairs which are the edges of MST, i.e. $\{A_i, A_j\} = E_{mst}$.
    **end if**
  **end for**

---

## 4 Experiments

### 4.1 Simulated dataset

In this experiment, the proposed method is compared with the state-of-the-art methods including FPC [37], ARD [28], HRRG [38], Crowd-BT [5], and Hodge-active [32]. A Monte Carlo simulation is conducted on 60 conditions (stimuli) whose scores are randomly selected from a uniform distribution on the interval of [1 5]. The assumptions are exactly the same with the experiment that we did in Section 3.3.3 and the observation error is set as 10%.

To obtain statistically reliable results, the simulation experiment is conducted 100 times. The relationship between the ground truth and the obtained estimated scores are evaluated by Kendall, PLCC, and the Root Mean Square Error (RMSE). Results are shown in Figure 3. It should be noted that as the PLCC, Kendall and RMSE values increase/decrease fast and look saturate when the trial number is large, it is difficult to visually distinguish the performances of different methods. Thus, in this paper, we rescale the Kendall and PLCC values by Fisher transformation, i.e. $y' = arctanh(y)$, and the RMSE value by function $y' = -\frac{1}{y}$.

**Qualitative analysis** Under the condition that each annotator has a 10% probability that inverses the vote, according to Figure 3, Hodge-active shows the strongest performance than others in ranking aggregation (Kendall) when the test budget (i.e. the number of comparisons) is small. With the increase of the trial number, the proposed Hybrid-MST method as well as the Crowd-BT shows comparable performance with Hodge-active. Regarding rating aggregation (PLCC and RMSE), the proposed Hybrid-MST method performs significantly better than the others except for that when the

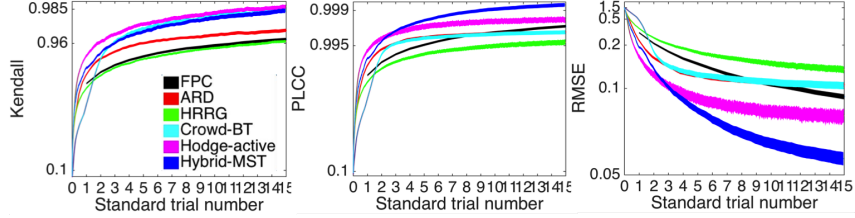

Figure 3: Monte Carlo simulation results. The color area represents 95% confidence intervals of the corresponding evaluated methods over 100 repetitions. For better visualization, the Kendall and PLCC are rescaled using Fisher transformation. RMSE is rescaled using $y' = -\frac{1}{y}$.

trial number is small, i.e. less than 2 or 3, the Hodge-active performs slightly better than Hybrid-MST. Crowd-BT shows similar performance with ARD in rating aggregation, which is lower than Hybrid-MST and Hodge-active but higher than HRRG.

**Saving budget compared to FPC** Following ITU-R BT.500 [39] and ITU-T P.910 [37], 15 standard trial number (i.e. 15 annotators to compare all $n(n-1)/2$ pairs) is the minimum requirement for FPC to generate reliable results. In this part, we compare how much budget can be saved by active sampling methods, i.e. Hybrid-MST, Hodge-active, and Crowd-BT. The mean of Kendall, PLCC and RMSE are used in a way that if $D$ pairwise comparisons in Hybrid-MST/Hodge-active/Crowd-BT could achieve the same precision as the FPC with 15 standard trial numbers, the saving budget $B_s$ is:

$$B_s = \left(1 - \frac{D}{\frac{n(n-1)}{2} \times 15}\right) \times 100\% \tag{11}$$

The obtained $B_s$ for Kendall, PLCC and RMSE are 77.11%, 74.89% and 74.89% for Hybrid-MST, and 84.57%, 68.61%, 71.65% for Hodge-active, respectively. Crowd-BT only has $B_s$ value for Kendall, which is 78.43%, as it needs more trial number to achieve the same FPC precision in PLCC and RMSE, which does not save budget.

**Computational cost** To evaluate the computational cost of each sampling method, the same Monte Carlo simulation test is conducted for $n = 10, 20$ and 100. The averaged time cost (milliseconds/pair) over 100 repetitions for each method is shown in Table 1. All computations are done using MATLAB R2014b on a MacBook Pro laptop, with 2.5GHz Intel Core i5, 8GB memory.

FPC is the simplest method without any learning process and therefore it is with the highest computationally efficiency. Besides, ARD, HRRG and Hodge-active also show their advantages in runtime. Crowd-BT shows similar runtime with our Hybrid-MST in GM mode. When Hybrid-MST is in MST mode, the runtime is approximately $n$ times more efficient than Crowd-BT and GM method. It should be noted that our proposed Hybrid-MST method only uses the GM method in the first standard trial (which can be easily reached in large-scale crowdsourcing labeling experiment) and then switches to the MST method, thus, in real application, our sampling strategy in most cases is in MST mode, which is much faster than Crowd-BT. Nevertheless, all runtimes are in a feasible range, even for large number of conditions and our unoptimized code (where the calculation of EIG for all pairs can be executed in parallel).

Table 1: Runtime comparison on simulated data (ms/pair)

| $n$ | FPC | ARD | HRRG | Crowd-BT | Hodge-active | Hybrid-MST | |
|-----|-----|-----|------|----------|--------------|------------|------|
| | | | | | | GM | MST |
| 10 | 0.11 | 1.24 | 0.38 | 85.69 | 0.34 | 48.72 | 6.16 |
| 20 | 0.10 | 0.62 | 0.34 | 188.56 | 0.22 | 153.61 | 8.97 |
| 100 | 0.10 | 0.16 | 0.65 | 3033.02 | 0.65 | 3007.08 | 30.04 |

To demonstrate the superiority of batch-mode sampling in real applications, we take a typical VQA experiment as an example (which also holds for player matching system, recommendation system, etc.). The typical presentation structure of sequential sampling methods (HRRG, Crowd-BT, Hodge-active, GM) for one pair comparison procedure is: pair presentation time ($T1$) + annotator's voting time ($T2$) + runtime of pairwise sampling algorithm ($T3$), where $T1$ and $T2$ are generally in total 15 seconds, $T3$ is determined by the used algorithm. Sequential sampling methods cannot generate

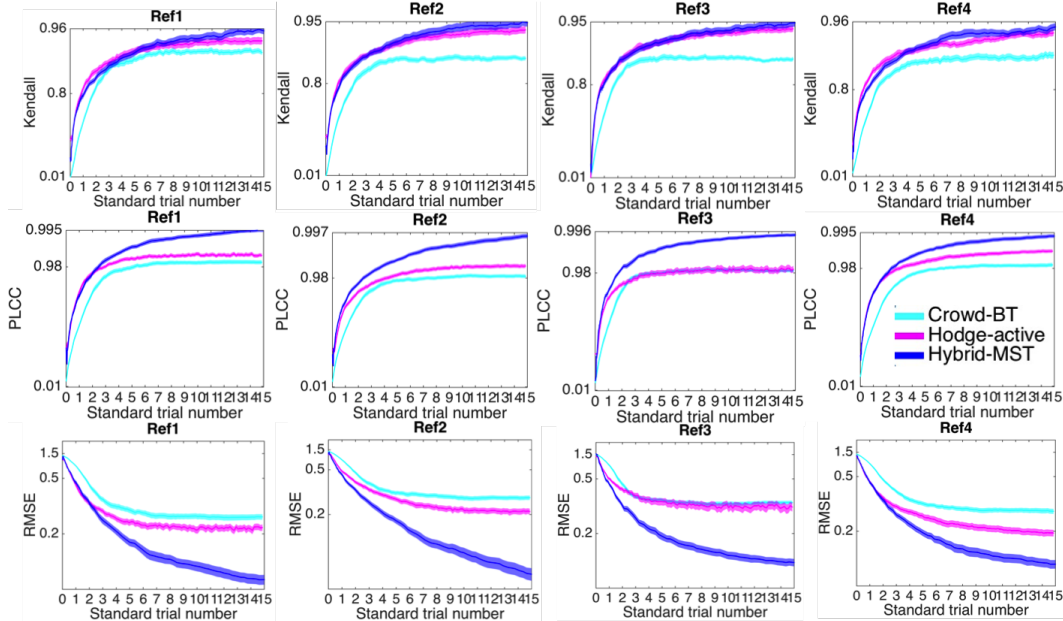

Figure 4: Performances of different sampling methods on VQA dataset. Color area represents 95% confidence intervals over 100 times iterations. For better visualization, Kendall and PLCC are rescaled using Fisher transformation. RMSE is rescaled using $y' = -\frac{1}{y}$.

a new optimal pair of objects to compare until the annotator is done with the previous pair. This introduces unacceptable delay in the system if multiple annotators work at the same time.

In contrast, the batch-based Hybrid-MST (in MST mode) can generate multiple pairs, which can be worked on in parallel by multiple annotators. Ideally (annotators work synchronously), the whole procedure for $n-1$ pairs needs $T1 + T2 + T3$ seconds. While in the worst case, the annotators work one after the other (just like in sequential method), which needs $T1 + T2 + T3$ seconds for only one pair. To make a comparison, the time cost of a whole pairwise comparison procedure including stimuli presentation time and voting time in a typical VQA experiment is shown in Table 2, which demonstrates that our method Hybrid-MST is particularly applicable in large-scale crowdsourcing experiment.

Table 2: Time cost (seconds) of comparing $n-1$ pairs in a typical VQA pair comparison experiment $(T1 + T2 + T3)$

| $n$ | Crowd-BT | Hodge-active | Hybrid-MST | | |
|---|---|---|---|---|---|
| | | | GM | MST(ideal case) | MST (the worst case) |
| 10 | 135.8 | 135.0 | 135.4 | 15.1 | 135.1 |
| 20 | 288.6 | 285.0 | 287.8 | 15.2 | 285.2 |
| 100 | 1782.0 | 1485.1 | 1782.0 | 17.9 | 1487.9 |

## 4.2 Real-world datasets

In this session, we compare our proposed Hybrid-MST with the state-of-the-art active learning methods, Crowd-BT [5] and Hodge-active [32]. For statistical reliability, each method is conducted 100 times. Two real-world datasets are used. Details are shown below.

**Video Quality Assessment(VQA) dataset**  This VQA dataset is a complete and balanced pairwise dataset from [38]. It contains 38400 pairwise comparisons for video quality assessment of 10 references from LIVE database [40]. Each reference contains 16 different types of distortions. 209 annotators attend this test.

**Image Quality Assessment (IQA) dataset**  This IQA dataset is a complete but imbalanced dataset from [26]. It contains 43266 pairwise comparison data for quality assessment of 15 references from

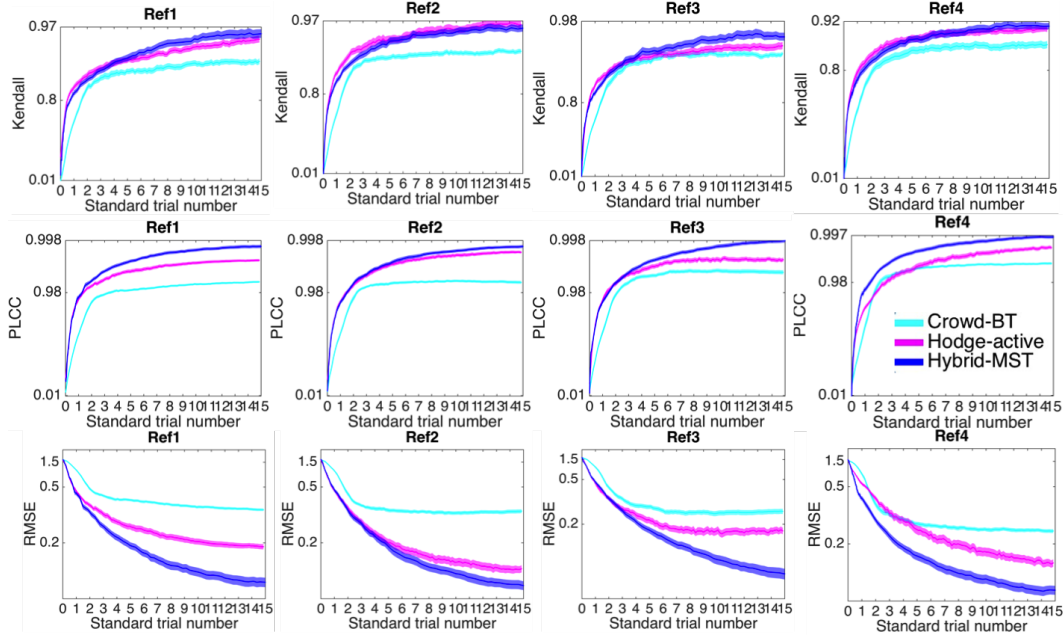

Figure 5: Performances of different sampling methods on IQA dataset. Color area represents 95% confidence intervals over 100 times iterations. For better visualization, Kendall and PLCC are rescaled using Fisher transformation. RMSE is rescaled using $y' = -\frac{1}{y}$.

LIVE 2008 [41] and IVC 2005 [42] database. Each reference contains 16 different types of distortions. 328 annotators from Internet attend the test.

As there is no ground truth for the real-world dataset, we consider the results obtained by all observers as ground truth. Again, Kendall, PLCC and RMSE are used as the evaluation methods. Due to the limitation of spaces, part of the results are shown in Figure 4 and 5.

In the real-world datasets where the annotator's labelings are much more noisy and diverse than our simulated condition, the proposed Hybrid-MST shows higher robustness to these noisy labelling than others. Regarding the ranking aggregation ability (Kendall), though Hodge-active still shows a bit stronger performance in ranking aggregation than Hybrid-MST when the trial number is small, it is not as much as in the simulated data. With the increase of the test budget, Hybrid-MST performs comparable or even better than Hodge-active. They both outperform Crowd-BT. Regarding the rating aggregation (PLCC and RMSE), Hybrid-MST always outperforms the others significantly. Hodge-active performs similar with Crowd-BT in VQA dataset, but much better than Crowd-BT in IQA dataset.

Both simulated and real-world experiments demonstrate that when the test budget is limited (2-3 standard trial numbers) and the objective is ranking aggregation, i.e. we care more about the rank order of the test candidates rather than their underlying scores, using Hodge-active is safer than Hybrid-MST. In all other conditions, Hybrid-MST is definitely more applicable considering both the aggregation accuracy and batch-mode execution.

## 5    Conclusions

In this paper, we present an active sampling strategy called **Hybrid-MST** for pairwise preference aggregation. We define the EIG based on local KLD where Bayes' theorem is adopted for finding the tractable computation form and Gaussian-Hermite quadrature is utilized for efficient estimation. Pair sampling is a hybrid strategy which takes advantages of both GM method and MST method, allowing for better ranking and rating aggregation in small and large trial number conditions. In both simulated experiment and the real-world VQA and IQA datasets, Hybrid-MST shows its outstanding aggregation ability. In addition, in crowdsourcing platform, the proposed batch-mode MST method could boost the pairwise comparison procedure significantly by parallel labeling.

## Footnotes

[1]Source code: https://github.com/jingnantes/hybrid-mst

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
