[Supplementary Material · Neurips2018_1781_supplementary.pdf]

# Hybrid-MST: A Hybrid Active Sampling Strategy for Pairwise Preference Aggregation (Supplementary materials)

**Jing Li**
LS2N/IPI Lab
University of Nantes
jingli.univ@gmail.com

**Rafal K. Mantiuk**
Computer Laboratory
University of Cambridge
rkm38@cam.ac.uk

**Junle Wang**
Turing Lab
Tencent Games
wangjunle@gmail.com

**Suiyi Ling, Patrick Le Callet**
LS2N/IPI Lab
University of Nantes
suiyi.ling, patrick.lecallet@univ-nantes.fr

## 1 Estimating covariance matrix by Hessian matrix

The MLEs $\hat{s}$ follow a multivariate Gaussian distribution. The covariance matrix of $\hat{\Sigma}$ could be estimated using the Hessian matrix of the $logL$, i.e.,

$$H = \begin{bmatrix} \frac{\partial^2 logL}{\partial s_1^2} & \cdots & \frac{\partial^2 logL}{\partial s_1 \partial s_n} \\ \cdots & \ddots & \cdots \\ \frac{\partial^2 logL}{\partial s_n \partial s_1} & \cdots & \frac{\partial^2 logL}{\partial s_n^2} \end{bmatrix} \tag{1}$$

Following [1][2], we construct a matrix C, which has the following form by augmenting the negative $H$ a column and a row vector of ones and a zero in the bottom right corner:

$$C = \begin{bmatrix} -\mathbf{H} & \mathbf{1} \\ \mathbf{1}' & 0 \end{bmatrix}^{-1} \tag{2}$$

The first $n$ columns and rows of $C$ form the estimated covariance matrix of $\hat{s}$, i.e., $\hat{\Sigma}$.

## 2 Simplification of Utility function

In our work, the EIG can be writen as:

$$U_{ij} = \int \sum_{y_{ij}} log \left\{ \frac{p(y_{ij}|s_{ij})}{p(y_{ij})} \right\} p(y_{ij}|s_{ij})p(s_{ij})ds_{ij} \tag{3}$$

In our study, $y_{ij}$ only has two values, 1 and 0. We define $p(y_{ij} = 1|s_{ij}) = p_{ij}$, and $p(y_{ij} = 0|s_{ij}) = q_{ij}$, thus, we have $p_{ij} = \frac{1}{1+e^{-s_{ij}}}$, $q_{ij} = 1 - p_{ij}$, then:

$$U_{ij} = \int \left( log \left\{ \frac{p_{ij}}{p(y_{ij} = 1)} \right\} p_{ij} + log \left\{ \frac{q_{ij}}{p(y_{ij} = 0)} \right\} q_{ij} \right) p(s_{ij})ds_{ij} \tag{4}$$

where

$$p(y_{ij} = 1) = \int p(y_{ij} = 1|s_{ij})p(s_{ij})ds_{ij}$$
$$= E(p(y_{ij} = 1|s_{ij})) \tag{5}$$
$$= E(p_{ij})$$

Similarly, we could obtain $p(y_{ij} = 0) = E(q_{ij})$. Thus, Equation 4 could be rewritten as:

$$U_{ij} = E(log(\frac{p_{ij}}{E(p_{ij})})p_{ij} + log(\frac{q_{ij}}{E(q_{ij})})q_{ij})$$
$$= E(p_{ij}log(p_{ij}) - p_{ij}logE(p_{ij})) + E(q_{ij}log(q_{ij}) - q_{ij}logE(q_{ij})) \tag{6}$$
$$= E(p_{ij}log(p_{ij})) - E(p_{ij})logE(p_{ij}) + E(q_{ij}log(q_{ij})) - E(q_{ij})logE(q_{ij})$$

# 3 Gaussian-Hermite quadrature estimation

In our paper,

$$E(p_{ij}log(p_{ij})) = \int p_{ij}log(p_{ij})p(s_{ij})ds_{ij}$$
$$= \int \frac{1}{1 + e^{-x}}log(\frac{1}{1 + e^{-x}})\frac{1}{\sqrt{2\pi}\sigma_{ij}}e^{-\frac{(x-(\hat{s}_i-\hat{s}_j))^2}{2\sigma_{ij}^2}}dx \tag{7}$$

Set $y = \frac{x-(\hat{s}_i-\hat{s}_j)}{\sqrt{2}\sigma_{ij}}$, thus, we have $x = \sqrt{2}\sigma_{ij}y + \hat{s}_i - \hat{s}_j$, Equation 7 could be rewritten as:

$$E(p_{ij}log(p_{ij}))$$
$$= \int \frac{1}{1 + e^{(-\sqrt{2}\sigma_{ij}y-(\hat{s}_i-\hat{s}_j))}}(-log(1 + e^{(-\sqrt{2}\sigma_{ij}y-(\hat{s}_i-\hat{s}_j))}))\frac{1}{\sqrt{\pi}}e^{-y^2}dy \tag{8}$$
$$= f(x)e^{-x^2}dx$$

where

$$f(x) = \frac{1}{1 + e^{(-\sqrt{2}\sigma_{ij}x-(\hat{s}_i-\hat{s}_j))}}(-log(1 + e^{(-\sqrt{2}\sigma_{ij}x-(\hat{s}_i-\hat{s}_j))}))\frac{1}{\sqrt{\pi}} \tag{9}$$

According to Gaussian-Hermite quadrature, the value of integrals with the form $\int_{-\infty}^{\infty} f(x)e^{-x^2}dx$ could be estimated by $\sum_{i=1}^{n} w_i f(x_i)$, where $n$ is the number of sample points used (please note that this $n$ is not the total number of objects in the paper), the $x_i$ are the roots of the physicists' version of the Hermite polynomial $H_n(x)(i = 1, 2, ..., n)$:

$$H_n(x) = (-1)^n e^{x^2}\frac{\partial^n}{\partial x^n}e^{-x^2} \tag{10}$$

and the associated weights $w_i$ are given by

$$w_i = \frac{2^{n-1}n!\sqrt{\pi}}{n^2[H_{n-1}(x_i)]^2} \tag{11}$$

In our study, n = 30.

# 4 Selection of rescale function for PLCC, Kendall and RMSE

With the increase of the trial number, the PLCC, Kendall and RMSE values increase/decrease rapidly and look saturate when the trial number is large. As shown in Figure 1(a)1(d), the difference of the performances of different methods is not readable. However, they are still distinguishable in the proper scales. Thus, we consider use Fisher transformation for PLCC and Kendall in this paper, i.e., $y' = arctanh(y)$, for better visualization as this transform could augment the difference when the value close to 1. The rescaled results are shown in Figure 1(b). Similar purpose of utilization of function $y' = -\frac{1}{y}$ for RMSE.

(a) Original cardinal scale on PLCC

(b) $y' = arctanh(y)$

(d) Original cardinal scale on RMSE

(e) $y' = -\frac{1}{y}$

Figure 1: Example of different re-scaling methods.

# 5 Complete results of Video Quality Assessment (VQA) datasets

Complete results of Kendall, PLCC and RMSE on Reference 5 - 10 of VQA dataset are shown in Figure 2, 3 and 4.

Figure 2: Kendall results on VQA dataset. Color area represents 95% confidence intervals over 100 times iterations. y-axis is rescaled using Fisher transformation.

# 6 Complete results of Image quality assessment (IQA) dataset

Complete results of Kendall, PLCC and RMSE on Reference 5 - 15 of IQA dataset are shown in Figure 5, 6 and 7.

Figure 3: PLCC results on VQA dataset. Color area represents 95% confidence intervals over 100 times iterations. y-axis is rescaled using Fisher transformation.

Figure 4: RMSE results on VQA dataset. Color area represents 95% confidence intervals over 100 times iterations. y-axis is rescaled using function $y' = -\frac{1}{y}$.

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

Figure 5: Kendall results on IQA dataset. Color area represents 95% confidence intervals over 100 times iterations. y-axis is rescaled using Fisher transformation.

Figure 6: PLCC results on IQA dataset. Color area represents 95% confidence intervals over 100 times iterations. y-axis is rescaled using Fisher transformation.

Figure 7: RMSE results on IQA dataset. Color area represents 95% confidence intervals over 100 times iterations. y-axis is rescaled using function $y' = -\frac{1}{y}$.