[Reviews · NeurIPS 2018]

Reviewer 1



This paper introduces a new approach to active learning for pairwise preference aggregation. This new approach, Hybrid-MST, appears to be competitive with some existing approaches and superior to others. Overall, the paper has two main weaknesses: - at times, it is fairly difficult to read - the interpretation of empirical results is not as clear-cut as it should In terms of paper organization, the introduction offers a good example of how the paper could be improved. Instead of a 1.5 page intro that includes a 1-page of literature review, this reviewer suggests the following alternative organization: - a 4-paragraph introduction that presents in an intuitive manner the why/what/how/main-results (WHY is this paper important? WHAT are you trying to solve (including one real-world motivating domain)? HOW do you solve the problem? in a nutshell, what are your MAIN RESULTS?). This structure would help the reader to go in depth in the remainder of the paper - a second section consisting in a traditional Related Work section Last but not least, it would also help to polish a bit the quality of the prose. In terms of the interpretation of the results, I would definitely like to see a discussion way beyond the paragraph between lines 247 and 253. When should one use Hybrid-MST and when Hodge-active? It would be great if the authors could provide more insights on this topic. Similarly, in Figure 3, in the left-most graph (Kendall), we have Hybrid-MST underperform Hodge, while in the mid-graph (PLCC) it is the other way around. What does this mean in terms of practical applicability? Why & when should we use one or the other?

Reviewer 2



Summary: This work aggregates noisy/inconsistent pairwise preferences into a rating for each item, as well as actively selecting the pairs to query. The technique used for aggregation is the Bradley-Terry model with computational saving techniques. The pairs are queried with Expected Information Gain (from the Bradley-Terry model) and either choosing the most informative pairs or choosing a batch of pairs corresponding to a MST built on the graph with edges based on the most informative pairs. Questions: Something that I didn’t quite understand is that this work claimed to run the preferences in batches, however, it doesn’t appear that they are run in batches for the first standard trial number. Can the authors please clarify this? The runtime for small problems (n=10-20) show that the algorithm runs relatively slowly and quadratically. How does this scale to the large real-world datasets? Why weren’t runtime experiments run for the real-world datasets? Quality: The techniques in this work seem to be relatively principled and fit into the related work well. The experiments show that this is an effective technique compared to the related work, at least for two datasets, IQA and VQA. Generally, in the experiments, the technique achieves the same results with half the number of comparisons. Clarity: This paper lays out concepts in an orderly way and is clear to read. One minor note is that the “Notation” section includes more than just notation, but modeling assumptions, so it is more than just notation. Also, I think the indentation of Algorithm 1 is off, the “end for” should come earlier. Originality: This work seems like a different take on a well-established problem. I’m not familiar enough with the preference aggregation literature to determine how similar this is to other work. Significance: I have some concerns with the runtime and efficiency of the method proposed in this work, however, it seems principled and has good empirical accuracies. After rebuttal: Thank you for the runtime experiments. I now see that the runtime is small compared to the annotation time.

Reviewer 3



This paper proposed a novel hybrid active strategy for collecting pair-wise samples for BT model estimation. The estimation is based on minimum spanning tree and several other insights. The proposed method shows reliable and more precise performance on simulated and real-world datasets. The paper is overall well organized. - missing some prior arts in the active learning and BLT model fields. Just to recall a few from top of my head. http://www.jmlr.org/papers/volume17/15-189/15-189.pdf http://proceedings.mlr.press/v28/wauthier13.pdf https://arxiv.org/abs/1109.3701 - There has been a lot of abbreviations in the paper which makes it difficult to read. - In table 1, the author claimed that the computation cost is comparable with Hodge based methods in large-scale crowd-sourcing experiments. It is not clear why this is a valid claim and is quite opposite from what stated in table 1. [based on the author response and the re-organized/commented computation cost table, it is clear that how the proposed approach gain advantage in terms of computation cost. It would be good to update the caption as well as the discussion in the paper draft to make this confusing point clear. In addition, since there are some "navie" non-active baselines, it would be good to clarify in the caption of table saying these approaches are not appropriate to compare and judge efficiency. ] Some additional comments: - In section 3.2, it would be better to clarify how the ground-truth labels/scores are assigned so it is easier to understand the RMSE/PLCC metrics. Though there is a space limits, some high-level description might help reader to understand in these two problems, what are the target problem and why intuitively ranking them are challenging tasks. - It seems a bit untypical that section 2.3.3 contains simulation while other experiments are in separate sections. Many of the simulation settings are discussed here which kind of complicated the reading process. - it would be better not to use many abbreviations in the main algorithm. Indeed, in Alg 1 the main steps of estimating info-gain eq(6) is not what used in the end. It is explicitly eq(7) and its practical approximations that enables the efficient computation. Highly suggest to make Alg. 1 referring to the exact algorithms used. -2.0 section is labeled as "notation" but actually contains key distribution assumptions. - Figure 1 (a) is almost unrecognizable. Please update it and include the parameters used to simulate these contour plots.